# Optimization of Spray-Drying Conditions of Microencapsulated Habanero Pepper (*Capsicum chinense* Jacq.) Extracts and Physicochemical Characterization of the Microcapsules

Kevin Alejandro Avilés-Betanzos [1], Juan Valerio Cauich-Rodríguez [2], Manuel Octavio Ramírez-Sucre [1] and Ingrid Mayanin Rodríguez-Buenfil [1,*]

1    Centro de Investigación y Asistencia en Tecnología y Diseño del Estado de Jalisco A.C., Subsede Sureste, Tablaje Catastral 31264, Km. 5.5 Carretera Sierra Papacal-Chuburná Puerto, Parque Científico Tecnológico de Yucatán, Mérida C.P. 97302, Yucatán, Mexico
2    Centro de Investigación Científica de Yucatán, Unidad de Materiales, Calle 43 No. 130 x 32 y 34, Colonia  Chuburná de Hidalgo, Mérida C.P. 97205, Yucatán, Mexico
*    Correspondence: irodriguez@ciatej.mx

**Abstract:** The Habanero pepper (*Capsicum chinense* Jacq.) is recognized worldwide for its unique organoleptic characteristics, as well as for its capsaicin content; however, other bioactive compounds, such as phenolic compounds with bioactive properties (mainly antioxidant capacity), have been extracted (ultrasound) and identified in this fruit. Moreover, the extracts obtained by ultrasound present a high sensitivity to environmental conditions, making spray drying a viable option to avoid the degradation of bioactive compounds while maintaining their properties after microencapsulation. Response surface methodology (RSM) has been used to optimize spray-drying conditions such as the inlet temperature (IT) and maltodextrin:extract (M:E) ratio. Thus, the objective of this work was to establish the optimal spray-drying conditions (IT and M:E) of a Habanero pepper extract with a final characterization of the spray-dried product. Results showed that the optimal spray-drying conditions included an IT = 148 °C with an M:E = 0.8:1 *w/w*, where the antioxidant capacity ($38.84 \pm 0.22\%$ inhibition), total polyphenol content ($6.64 \pm 0.08$ mg Gallic acid equivalent/100 g powder), and several individual polyphenols, such as Protocatechuic acid (26.97 mg/100 g powder), Coumaric acid (2.68 mg/100 g powder), Rutin (18.01 mg/100 g powder), Diosmetin (1.74 mg/100 g powder), and Naringenin (0.98 mg/100 g powder), were evaluated. The microcapsules showed a spherical shape with concavities and moisture less than 5%, and the inclusion of bioactive compounds was confirmed using UPLC and FTIR. The final dried product has the potential to be used as an ingredient for functional food development.

**Keywords:** optimization; spray drying; *Capsicum chinense*; antioxidant capacity; polyphenols

## 1. Introduction

The Habanero pepper (*Capsicum chinense* Jacq.) is an herbaceous plant within the Solanaceae family; this fruit has a high commercial value in Mexico and internationally. It presents unique organoleptic characteristics with high contents of important bioactive compounds, highlighting capsaicin and polyphenols, as well as the antioxidant capacity as important bioactive properties [1,2].

This fruit is representative of the Yucatan Peninsula region in Mexico since it exhibits 22.4% of the national production, representing a significant impact on this geographical area [3]. Therefore, in 2010, the denomination of origin "Chile Habanero de la Península de Yucatán" was granted [4]. This denomination has aroused the interest of the pharmaceutical, food, and cosmetic industries on other secondary metabolites, different from Capsaicinoids, contained in the Habanero pepper, such as polyphenols and other bioactive compounds

developed within the shikimic acid and metabolic pathways of phenylpropanoids, synthesized by plants as a defense mechanism in response to biotic and abiotic stress [5,6]. Moreover, these metabolites present different properties such as anti-inflammatory, antidiabetic, antilipogenic, and antioxidant capacities when included in diets, providing benefits to human health [7,8]. Thus, the search for obtaining these bioactive molecules has led to the application of different extraction methodologies. Ultrasound-assisted extraction (UAE) is one of the most promising due to its low extraction time, little use of toxic solvents, and high yield in comparison to other methodologies such as maceration or Soxhlet. In this way, UAE is a green extraction option for the phenolic compounds from the Habanero pepper [9]. For instance, Oney-Montalvo et al. [10] reported the extraction of phenolic compounds from immature Habanero peppers (green) using 80% methanol as the solvent with a sonic bath (130 W, 42 kHz, 30 min), obtaining concentrations within the range from 39.61 mg to 79.34 mg Gallic acid equivalent/100 g dry pepper. *Capsicum chinense* Jacq. extracts at the mature state (orange color) have also been reported with concentrations up to $208.42 \pm 31.01$ mg Gallic acid equivalent/100 g dry pepper when simultaneously implementing UAE (42 kHz) and methanol as the solvent [11].

Although the Habanero pepper extracts have high concentrations of polyphenols, these are susceptible to rapid degradation due to environmental conditions, such as Ultra Violet light, moisture, high temperature, pH, oxygen, among others [12,13]. Using spray drying is a viable option for increasing the shelf life of bioactive compounds found in Habanero peppers, as well as preserving their bioactive properties through the use of natural coating or encapsulating agents such as polysaccharides, proteins, and/or gums. By using a suitable proportion of these natural products (encapsulating agents) with Habanero pepper extract and establishing optimal injection conditions at the drying chamber, microcapsules containing the desired metabolite can be obtained. This prevents interaction with harsh environmental conditions and limits degradation [14,15].

Even though spray drying is a widely used microencapsulation technique for encapsulating phenolic compounds extracted from various food matrices, such as blueberry [16], grape [17], chokeberry [18], pomegranate [19], and acerola [20], it has not been yet used for encapsulating phenolic compounds from plants of the *capsicum* genus [21–25]. Therefore, in order to minimize the loss of metabolites and antioxidant capacity in the microencapsulates obtained from the Habanero pepper extract, it is imperative to employ statistical tools to optimize the operating conditions of the spray dryer. This includes considering the conditions of the microencapsulation process, such as the inlet temperature (IT) and the amount (ratio) of encapsulation agent used [26,27]. Response surface methodology (RSM) plays a critical role in achieving the optimal conditions for spray drying [28]. The application of RSM in spray drying has proven to be effective in minimizing variability and improving the quality of the final product, making it an essential tool for the food industry [29]. To achieve the optimization of spray-drying conditions, RSM involves a series of mathematical and statistical techniques where empirical models are adjusted to the data obtained from the experimental design. This is achieved through the use of linear or quadratic polynomial functions, followed by modeling and optimization techniques to analyze and improve the experimental conditions through the use of statistical software [30]. Once the preliminary experimental phase is completed, the linearity of the data must be analyzed; generally, the data obtained do fit a linear model, which indicates that there is no curvature in the data and it is in an area far from optimal; therefore, new experiments must be carried out (stepped method) to improve the desired response variables, which translates into a fit of the data to a quadratic model. If the data do not fit a linear model, it would only be necessary to add the star points and analyze the results together with those of the first experimental design to establish the fit to a quadratic model [29]. An advantage of using RSM, in addition to the cost savings from reducing the number of experiments, is its ability to generate a mathematical model that can predict the behavior of response variables, such as the concentration of microencapsulated bioactive compounds, based on various

spray-drying conditions, including inlet temperature, encapsulating agent ratio, injection flow, outlet temperature, among others [29,31].

To date, no information has been found regarding the optimization of microencapsulation conditions of phenolic compounds found in the Habanero pepper fruit nor their antioxidant capacity in microencapsulates. Therefore, the objective of this study was to optimize the spray-drying conditions, evaluate the antioxidant capacity and phenolic compounds present in the microencapsulates, and provide a comprehensive characterization of the resulting product.

## 2. Materials and Methods

### 2.1. Raw Materials

Habanero pepper (*Capsicum chinense* Jacq.) was grown under greenhouse conditions in the community of Chicxulub pueblo, Yucatán, Mexico with geographic coordinates 21°08′50.5″ N, 89°29′42.8″ W used.

The Habanero pepper was cultivated in a lithic leptosol soil according to the classification of the "World Reference Base for soil resources (WRB)", known as Tzek'el lu'um by the Mayan classification. Fruits were harvested on 11 December 2019 in an immature state (green color) three months after planting.

### 2.2. Habanero Pepper Drying Process

The freshly harvested Habanero pepper in an immature state (green color) was transferred to the CIATEJ facilities (southeast campus), where a classification of the product was carried out, setting apart the fruits in an immature state (green color) from the fruits that presented changes in color (green–orange and orange color) as well as from other residues such as leaves, stems, and peduncles.

The collected and classified Habanero pepper fruits were placed in aluminum trays and dried with a FELISA oven (model FE-292) at a temperature of 65 °C for 72 h [32]. The dried Habanero pepper fruits were ground with an Oster® blender (México city, México) and sieved with a #35 sieve (500 μm particle size); finally, the Habanero pepper powder was placed in plastic bags lined with aluminum foil and stored at room temperature until use.

### 2.3. Habanero Pepper Polyphenols Extraction

The extraction process was carried out according to Oney-Montalvo et al. [32] with some modifications. A total of 25 g of Habanero pepper powder was added to 125 mL of a methanol:water solution (80:20) and then sonicated (water bath) for 30 min at a frequency of 42 kHz (BRANSON® model 351).

The extract obtained was centrifuged for 30 min at 4 °C and 4700 rpm; the resulting supernatant was filtered with a nylon filter (0.2 μm) and weighted with the help of an OHAUS® (model PA224C) balance. Finally, it was cooled in a fridge (10 °C) for less than one hour until its microencapsulation. Aliquots of the filtered extract were taken and analyzed for antioxidant capacity (Ax), total polyphenol content (TPC), polyphenol profile (PP), and total Capsaicinoids (TCA) determination.

### 2.4. Microencapsulation of the Habanero Pepper Extract

Experimental Design

Response surface methodology (RSM) was used to optimize the microencapsulation conditions studying its effects on the Habanero pepper extract microcapsules. A Central Composite Design (CCD) $2^2$ was employed [33]. The CCD consisted of two main factors with two levels each ($\pm 1$); the factors were maltodextrin:extract ratio (1:1–1:2 *w/w*) and inlet temperature (100–140 °C).

The central points, which make the model obtained more accurate, were the average distance between the highest and lowest value of the factor to be evaluated, and were established at an inlet temperature of 120 °C and a 1.5:1 *w/w* ratio of maltodextrin:extract. The star points (second experimental design) were established at an IT = 92 °C ($-1.414$)

and 148 °C (1.414) and a maltodextrin:extract (*w/w*) ratio M:E = 0.8:1 *w/w* (−1.414) and
2.2:1 *w/w* (1.414). Values are shown in Table 1.

**Table 1.** Central Composite Design $2^2$ for the optimization of spray-drying parameters in the microencapsulation process of Habanero pepper extract with high antioxidant capacity.

| Exp | Coded Values | | Real Values | | Response Variables | |
|-----|-------|-------|------------|--------------|----------------------|-----------------------------|
| | $X_1$ | $X_2$ | IT (°C) | M:E (*w/w*) | Ax (% Inhibition) | TPC (GAE mg/100 g PW) |
| 1 | −1 | −1 | 100 | 1 | $Y_1$ | $Z_1$ |
| 2 | 1 | −1 | 140 | 1 | $Y_2$ | $Z_2$ |
| 3 | −1 | 1 | 100 | 2 | $Y_3$ | $Z_3$ |
| 4 | 1 | 1 | 140 | 2 | $Y_4$ | $Z_4$ |
| 5 | 0 | 0 | 120 | 1.5 | $Y_5$ | $Z_5$ |
| 6 | 0 | 0 | 120 | 1.5 | $Y_6$ | $Z_6$ |
| 7 | 0 | 0 | 120 | 1.5 | $Y_7$ | $Z_7$ |
| 8 | 0 | 0 | 120 | 1.5 | $Y_8$ | $Z_8$ |
| 9 | −1.414 | 0 | 92 | 1.5:1 | $Y_9$ | $Z_9$ |
| 10 | 1.414 | 0 | 148 | 1.5:1 | $Y_{10}$ | $Z_{10}$ |
| 11 | 0 | −1.414 | 120 | 0.8:1 | $Y_{11}$ | $Z_{11}$ |
| 12 | 0 | 1.414 | 120 | 2.2:1 | $Y_{12}$ | $Z_{12}$ |

Note: Exp = Experiment; IT = inlet temperature; M:E = ratio of maltodextrin gram per 1 g of Habanero pepper extract; *w/w* = weight/weight; Ax = antioxidant capacity; TPC = total phenolic content; GAE = Gallic acid equivalent; PW = powder.

### 2.5. Spray-Drying Process

The spray-drying process was conducted according to Chong et al. [27] with some modifications. The microencapsulation process began by weighing Habanero pepper (*Capsicum chinense* Jacq.) extract. Based on the obtained weight, the amount of maltodextrin (DE20, Maltrin® M200, Mexico City, Mexico) was added, according to experimental design.

The maltodextrin was then diluted under continuous stirring (heating plate and magnetic stirrer) with distilled water at room temperature until a homogeneous solution was achieved. Then, the Habanero pepper extract (previously weighed) was added to the maltodextrin solution with continuous stirring until a homogeneous solution was accomplished.

The homogeneous solution of maltodextrin and Habanero pepper extract was finally injected into the spray dryer (MOBILE MINOR™ GEA® Model MM standard, Düsseldorf, Germany) with a 10 mL/min flow achieved by a peristaltic pump (WATSON MARLOW®, model 520S, Düsseldorf, Germany). The injection was maintained with an air pressure of 3.5 Bar (22,000 rpm) to keep atomizer rotating. A constant flow of hot air (temperature according to experimental design) of 80 kg/h was established inside the drying chamber. The microencapsulates were collected with 1 l specialized bottles (GEA®, Düsseldorf, Germany). The powder was stored at room temperature, labeled, and lined with aluminum foil until use.

### 2.6. Determination of the Total Polyphenol Content in Habanero Pepper Extracts and Microencapsulates

The determination of the total polyphenol content (TPC) in the Habanero pepper extracts was carried out using the Folin–Ciocalteu methodology according to Oney-Montalvo et al. [32].

An aliquot (25 µL) of Habanero pepper extract was taken. The extract was diluted with distilled water (1:1 ratio). Finally, 3 mL of water and 250 µL of Folin–Ciocalteu reagent (St. Louis, MO, USA) were added before incubation for 5 min. After incubation, 750 µL of sodium carbonate ($Na_2CO_3$, 20%, St. Louis, MO, USA) and 950 µL of distilled water were added; then, the final solution was incubated for 30 min. Finally, the readings were

conducted at a wavelength of 765 nm using a quartz cell and a JENWAY® (model 6715 Ultraviolet–visible light, IL, USA) spectrophotometer. The results were presented as Gallic acid equivalents in mg of Gallic acid equivalent per 100 g of dry Habanero pepper (GAE mg/100 g dry Habanero pepper), according to the calibration curve (Figure S1).

To carry out the microencapsulated Habanero TPC assess, microcapsules were washed and reconstituted. First, 500 mg of powder was washed with 2.5 mL of methanol and stirred (vortex mixer Maxi Mix® II, MA, USA) until a homogeneous mix was observed (<5 min). Then, the mix was centrifuged during 20 min at 4700 rpm and 4 °C, and the obtained supernatant was discarded to remove non-microencapsulated polyphenols [27]. A volume of 2.5 mL of distilled water was added to washed and stirred (vortex mixer Maxi Mix® II) microcapsules until diluted. This solution was filtered (0.2 μm nylon filter) and used to determine the microencapsulated total polyphenol content. Results were reported as mg GAE/100 g of powder, as described before.

### 2.7. Polyphenol Profile Determination in Habanero Pepper Extracts and Microencapsulates

The quantification of the individual polyphenols was carried out with a UPLC Acquity H Class ultra-pressure chromatograph (Waters, Milford, MA, USA) with a detector of diode array (DAD) and an Acquity HSS C18 reverse phase column (Waters, Milford, MA, USA).

The methodology described by Oney-Montalvo et al. [10] was applied, using a flow rate of 0.5 mL/min, 0.2% acetic acid as the mobile phase A, and acetonitrile with acetic acid at 0.1% as mobile phase B. The calibration curve was prepared with 16 polyphenol standards (Gallic acid, Protocatechuic acid, Chlorogenic acid, Coumaric acid, Cinnamic acid, Vanillic acid, Ferulic acid, Ellagic acid, Rutin, Quercetin, Luteolin, Kaempferol, Vanillin, Naringenin, and Diosmetin) purchased from Sigma-Aldrich® (St. Louis, MO, USA). A stock solution was made with a concentration of 1 mg/mL to prepare a curve in the range from 1 to 75 μg/mL.

The polyphenols were identified in the samples by comparing them with the retention time of the standards. The standards chromatograms are shown in Figure S2. Results were reported as mg/100 g dried pepper (extract) and mg/100 g powder (microencapsule).

### 2.8. Total Capsaicinoids Determination in Habanero Pepper Extract and Microencapsules

The quantification of the individual Capsaicinoids was carried out with a UPLC Acquity H Class ultra-pressure chromatograph (Waters, Milford, MA, USA), as described in Section 2.8.

In order to determine the Capsaicinoid content, a calibration curve was made by weighting 1 mg capsaicin (Lot#SLCF5039) and 0.5 mg dihydrocapsaicin (Lot#BCCB2256) of standards (Sigma-Aldrich®, St. Louis, MO, USA), grading up to 1 mL with acetonitrile (stock 1). From stock 1, 200 μL was taken, grading with acetonitrile:water up to 2 mL (stock 2).

The curve was established in a range from 5 to 80 μg/mL for capsaicin and from 2.5 to 40 μg/mL for dihydrocapsaicin.

Determination of total Capsaicinoids, in both extract and microencapsulates, was made according to Chel-Guerrero et al. [34] with two mobile phases; acetonitrile was established as phase A and a 0.1% formic acid solution as phase B, at 0.5 mL/min flow rate in a constant ratio (60:40) during the injection (5 min). Results were reported as mg/g dried pepper (extract) and mg/g powder (microencapsulate).

### 2.9. Antioxidant Capacity in Habanero Pepper Extract and Microencapsules

The DPPH radical scavenging methodology was used according to Chel-Guerrero et al. [34] to determine the antioxidant capacity. The 2,2-diphenyl-1-picrylhydrazyl (DPPH) reagent of the Sigma Aldrich® brand (D9132 -1G, St. Louis, MO, USA) was weighted (3.3 mg); then, reagent grade methanol was added (100 mL). The obtained DPPH-solution was adjusted (absorbance 0.700 ± 0.002) with a spectrophotometer JENWAY® (model 6715 Ultraviolet–visible light, IL, USA) reading at 515 nm.

To establish the antioxidant capacity, a 100 µL volume of sample (Extract or reconstituted microcapsule) were taken, then a 3.9 mL volume of DPPH-adjusted solution was added. The mixed solution was stirred and incubated for 30 min. Finally, the samples were read at 515 nm.

The results of the extract and microencapsulates were presented as percentage of inhibition (%), according to Equation (1):

$$\% \ Inhibition = \left( \frac{DPPH_{adj} - DPPH_{sam}}{DPPH_{adj}} \right) \tag{1}$$

$DPPH_{adj}$ = Absorbance of adjusted *DPPH* solution.
$DPPH_{sam}$ = Absorbance of sample (extract or microcapsule reconstituted).

### 2.10. Physicochemical Characterization of Habanero Pepper (Capsicum chinense Jacq.) Extract Microencapsulated Obtained by Spray Drying

#### 2.10.1. Morphology Using Scanning Electron Microscope (SEM)

The SEM analysis provided images of morphology and approximate size of the Habanero pepper extract microcapsules. The microcapsule physical properties were determined with a JEOL 6360 LV scanning electron microscope (Tokyo, Japan). Samples (maltodextrin DE20, spray-dried maltodextrin DE20, and Habanero pepper extract microencapsulates) were gold-plated and then observed at 25 kV accelerating voltage, aperture 1, and spot size 31. Particle dimensions were determined using ISIS JEOL SEM software (version 6.10), according to Barbieri et al. [35].

#### 2.10.2. Fourier Transform Infrared Spectroscopy (FTIR)

A Nicolet 8700 FT-IR spectrometer (Thermo Scientific, Madison, WI, USA) equipped with an attenuated total reflectance (ATR) accessory using germanium glass was used to obtain FTIR spectra of the samples (Habanero pepper extract, maltodextrin DE20, and Habanero pepper extract microencapsulated). Scans were made in the spectral range between 4000 and 650 cm$^{-1}$ and were recorded after averaging 100 scans with a 2 cm$^{-1}$ resolution [35].

#### 2.10.3. Colorimetry

Color of samples was evaluated with a colorimeter ColorMeter Pro$^{®}$ (Software version 2.1.18, Guangdong, China) according to Siacor et al. [36] with some modifications. Five grams of sample of each microencapsulate was used to cover the entire base (reading area) of the quartz sample cup (HunterLab$^{®}$ model 04-7209-00, VA, USA) while a white background was placed at a height of 25 mm. Prior to the readings, the equipment was calibrated according to the manufacturer instructions to later place the sample and read the color by triplicate.

Results were reported according to the CIELAB scale (*L\**, *a\**, *b\**); Chroma and Hue° values were obtained using Equations (2) and (3):

$$Chroma = \left( a^{*2} \times b^{*2} \right)^{1/2} \tag{2}$$

$$Hue^{\circ} = \arctan\left( \frac{b^{*}}{a^{*}} \right) \tag{3}$$

#### 2.10.4. Moisture

Moisture contents of Habanero pepper and microencapsulates were analyzed according to the methodology reported by Li et al. and Tolun et al. [37,38] with some modifications. With an OHAUS$^{®}$ thermobalance (model MB90, NJ, USA), the moisture content was determined by placing 0.5 g sample in an aluminum plate; the samples were kept at a constant temperature of 105 °C until a constant weight was reached (weight loss < 1 mg in 60 s).

The moisture content was reported in percentage (%) according to the lost weight of the sample by triplicate.

### 2.11. Statistical Analysis

Experiments were performed randomly. Data presented were reported as means $\pm$ standard deviations. Data analysis, first- and second-order model fit analysis, canonical analysis, and regression coefficients were performed using the statistical software Statgraphics Centurion XVII.II-X64 (Statgraphics Technologies Inc., Virgin, UT, USA).

## 3. Results

### 3.1. Physicochemical Characteristics of the Habanero Pepper Fruit

Habanero Pepper Fruit Moisture

The moisture percentage of the Habanero pepper grown under greenhouse conditions in the Tzek'el soil type reported a value of 83.19 $\pm$ 0.40%, lower than that reported by Vásquez-Velázquez et al. [39] for immature Habanero peppers (*Capsicum chinense* Jacq.) (91 $\pm$ 0.00%). This could be the effect of the type of soil since a stony soil (Tzek'el lu'um, Mayan classification) classified as leptosol was used for this work. This soil presented a high percolation rate (rapid water filtration) as well as difficulty for root development, thus decreasing the water supply [40–42].

### 3.2. Optimization of the Spray-Drying Conditions

3.2.1. TPC and Ax of the Habanero Pepper Extract

Prior to carrying out the treatments by the optimization design, antioxidant capacity (Ax) and total polyphenol content (TPC) of the extract of the Habanero pepper (*Capsicum chinense* Jacq.) were analyzed. The Habanero pepper extract presented an antioxidant capacity of 95.03 $\pm$ 0.08% inhibition and 32.97 $\pm$ 0.08 mg GAE/100 g dried pepper.

3.2.2. TPC and Ax from Experimental Design

According to the Central Composite Design $2^2$ with four central points, eight experiments were carried out randomly at different conditions of inlet temperature (100 °C, 120 °C, and 140 °C) and ratios of maltodextrin gram per gram of extract (1:1, 1.5:1, and 2:1 $w/w$, maltodextrin:extract). Microencapsulates obtained in each treatment were analyzed by triplicate to determine the antioxidant capacity and the total polyphenol content.

The first-order analysis (Table S1) for these eight experiments showed that the antioxidant capacity did not fit a linear model with a $p$-value = 0.0671 and $R^2$ = 29.5; in contrast, TPC did fit a linear model ($p$-value < 0.05). The lack of adjustment evidenced the presence of a curvature of the data, an approximation to the optimal values, allowing the addition of star points and obtaining a second experimental design.

According to RSM, the star points were added (experiments from nine to twelve, second experimental design) with the coded values 0 and $\pm\sqrt{2}$ ($\pm 1.414$). Results are shown in Table 2.

An inlet temperature condition of 140 °C and a ratio of maltodextrin gram per gram of extract of 1:1 $w/w$ showed the highest TPC (7.75 $\pm$ 0.25 mg GAE/100 g PW) in the microencapsulated Habanero pepper extract; whereas, the lowest TPC (2.89 $\pm$ 0.22 mg GAE/100 g PW) was obtained under 100 °C (inlet temperature) and 2:1 $w/w$ (M:E) conditions.

The higher antioxidant capacity (31.43 $\pm$ 0.22% inhibition) was obtained when an inlet temperature of 120 °C and an encapsulating agent ratio of 0.8:1 $w/w$ (maltodextrin:extract) were implemented; the lowest Ax (2.05 $\pm$ 0.36% inhibition) was observed under the central point condition (120 °C, 1:1.5 $w/w$). No linear correlation was detected between Ax and TPC ($R^2$ < 0.1).

**Table 2.** Values of antioxidant capacity and total polyphenol content obtained from the $2^2$ Central Composite Design for the spray-drying condition optimization of a Habanero pepper extract.

| | Factors | | | | Response Variables | |
| --- | --- | --- | --- | --- | --- | --- |
| | Coded Values | | Real Values | | Ax (% Inhibition) | TPC (mg GAE/100 g Powder) |
| Exp | $X_1$ | $X_2$ | IT (°C) | M:E (*w/w*) | | |
| 1 | −1 | −1 | 100 | 1:1 | 25.71 ± 0.14 [j] | 5.67 ± 0.14 [de] |
| 2 | 1 | −1 | 140 | 1:1 | 14.29 ± 0.25 [g] | **7.75 ± 0.25 [h]** |
| 3 | −1 | 1 | 100 | 2:1 | 14.52 ± 0.22 [g] | 2.89 ± 0.22 [a] |
| 4 | 1 | 1 | 140 | 2:1 | 12.29 ± 0.14 [f] | 3.05 ± 0.14 [a] |
| 5 | 0 | 0 | 120 | 1.5:1 | 6.21 ± 0.08 [c] | 2.98 ± 0.83 [a] |
| 6 | 0 | 0 | 120 | 1.5:1 | 2.05 ± 0.36 [a] | 4.61 ± 0.36 [b] |
| 7 | 0 | 0 | 120 | 1.5:1 | 4.92 ± 0.08 [b] | 6.11 ± 0.08 [f] |
| 8 | 0 | 0 | 120 | 1.5:1 | 11.32 ± 0.43 [d] | 6.09 ± 0.43 [f] |
| 9 | −1.414 | 0 | 92 | 1.5:1 | 11.78 ± 0.22 [e] | 5.40 ± 0.00 [cd] |
| 10 | 1.414 | 0 | 148 | 1.5:1 | 23.99 ± 0.08 [i] | 5.23 ± 0.48 [c] |
| 11 | 0 | −1.414 | 120 | 0.8:1 | **31.43 ± 0.22 [k]** | 6.56 ± 0.15 [g] |
| 12 | 0 | 1.414 | 120 | 2.2:1 | 17.36 ± 0.16 [h] | 5.91 ± 0.12 [ef] |

Note: Exp = Experiment; IT = inlet temperature; M:E = ratio of maltodextrin per gram of Habanero pepper extract; *w/w* = weight/weight ratio; Ax = antioxidant capacity; TPC = total polyphenol content; GAE = Gallic acid equivalent; values are means ± SD (*n* = 3). Different letters in the same column indicate a significant statistical difference.

### 3.2.3. Modeling of the Spray-Drying Conditions for Microencapsulated Antioxidant Capacity

The second-order analysis of the completed experimental design (twelve experiments: first eight experiments plus star points) of the antioxidant capacity showed a *p*-value = 0.001 and an $R^2$ = 74.6, indicating an adjustment of the antioxidant capacity values to a second-order model. The analysis involved obtaining multiple regression coefficients (Table S2), which were then used to design a prediction Equation (4) for the antioxidant capacity of the microencapsulated Habanero pepper (Capsicum chinense Jacq.) extract, as follows:

$$Y = 301.59 - 3.17846\ X_1 - 133.013\ X_2 + 0.0118935\ X_1{}^2 + 0.23025\ X_1 X_2 + 32.3595\ X_2{}^2 \quad (4)$$

Y = Antioxidant capacity (% inhibition).
$X_1$ = Inlet temperature (°C).
$X_2$ = Maltodextrin gram ratio per gram of extract (maltodextrin:extract, *w/w*).

According to the mathematical model, to obtain an optimal antioxidant capacity in the microencapsulation of up to 41.1% inhibition, an inlet temperature of 92 °C and a 0.8 g maltodextrin ratio per 1 g of extract should be implemented. Figure 1 shows the response surface (a) and the contour plot (b) obtained from the modeling of the antioxidant capacity of the microencapsulated Habanero pepper (*Capsicum chinense* Jacq.) extract. The surface plot shows the behavior of the Ax of the microencapsulates, where a plateau of minima (blue color) represents the lowest antioxidant capacity and the surroundings (red color) represent the highest antioxidant capacity. In Figure 1b, the intersection of the spray-drying conditions (IT = 92 °C, M:E = 0.8:1 *w/w*) to achieve the optimal response (optimal antioxidant capacity) is shown with the symbol "+".

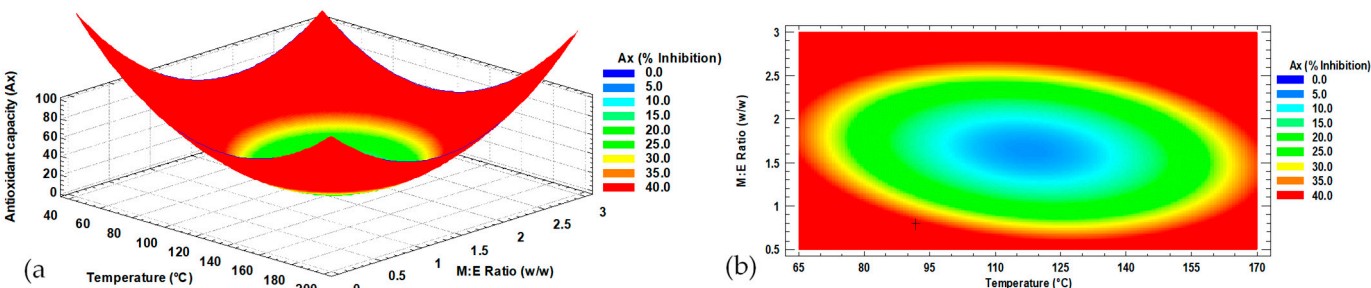

**Figure 1.** Antioxidant capacity response surface (**a**) and contour (**b**) plots with factors of input inlet temperature and maltodextrin ratio. Ax = Antioxidant capacity; M:E = maltodextrin ratio per gram of Habanero pepper extract; $w/w$ = weight/weight.

The maltodextrin ratio ($p = 0.00$) showed an effect on the antioxidant capacity of the microencapsulated extract; meanwhile, the temperature ($p = 0.61$) and interaction ($p = 0.05$) of both main factors did not show an effect on this response variable.

### 3.2.4. Modeling of Spray-Drying Conditions

According to the Central Composite Design with star points, TPC fitted to a second-order model ($p$-value = 0.00, $R^2$ = 50.54); whereas, for the polyphenol profile, only Protocatechuic acid, Coumaric acid, Rutin, Diosmetin, and Naringenin fitted to a second-order model (Table 3).

**Table 3.** Results of multiple regression analysis of individual polyphenols of a Habanero pepper extract microencapsulated by spray drying.

| Metabolite | $p$-Value | $R^2$ |
|---|---|---|
| Protocatechuic acid | 0.0000 | 77.20 |
| Coumaric acid | 0.0000 | 83.94 |
| Rutin | 0.0000 | 85.91 |
| Diosmetin | 0.0000 | 83.39 |
| Naringenin | 0.0000 | 72.22 |

The regression coefficients were used to design the prediction equations of the TPC and the concentration of each microencapsulated individual polyphenol; the equations are shown in Table 4.

**Table 4.** Prediction equation for TPC and for concentration of microencapsulated individual polyphenols.

| Metabolite | Equation |
|---|---|
| TPC | $Y = -0.464268 + 0.120413\,X_1 - 1.29686\,X_2 - 0.00014918\,X_1^2 - 0.048\,X_1X_2 + 1.64567\,X_2^2$ |
| Protocatechuic acid | $Y = 16.5604 + 0.996392X_1 - 72.6487X_2 - 0.00479653X_1^2 + 0.0630083X_1X_2 + 17.4266\,X_2^2$ |
| Coumaric acid | $Y = 14.4128 - 0.141818\,X_1 - 5.68653\,X_2 + 0.000530469\,X_1^2 + 0.00554167X_1X_2 + 1.46141\,X_2^2$ |
| Rutin | $Y = 123.536 - 1.26763\,X_1 - 50.1663\,X_2 + 0.00394197\,X_1^2 + 0.163892\,X_1X_2 + 8.6508\,X_2^2$ |
| Diosmetin | $Y = 11.268 - 0.15616\,X_1 - 1.78018\,X_2 + 0.00076042\,X_1^2 - 0.0211333\,X_1X_2 + 1.33133\,X_2^2$ |
| Naringenin | $Y = 3.07087 - 0.0534309\,X_1 - 0.270552\,X_2 + 0.000324066\,X_1^2 - 0.0119667\,X_1X_2 + 0.548494\,X_2^2$ |

Note: TPC = Total polyphenol content; Y = TPC (mg GAE/100 g powder) or metabolite (mg/100 g powder); $X_1$ = inlet temperature (°C); $X_2$ = maltodextrin ratio per gram of extract.

Table 5 shows the values of the optimal spray-drying predicted conditions as well as the optimal predicted value for TPC and each individual polyphenol microencapsulate according to the canonical analysis.

**Table 5.** Spray-drying predicted conditions to obtain an optimal predicted concentration of microencapsulated metabolites of a Habanero pepper extract.

| Metabolite | Spray-Drying Conditions | | Optimal Response (mg/100 g Powder) |
|---|---|---|---|
| | M:E (*w/w*) | IT (°C) | |
| TPC | 0.8:1 | 148 | 8.42 * |
| Protocatechuic acid | 0.8:1 | 109 | 26.97 |
| Coumaric acid | 0.8:1 | 92 | 2.68 |
| Rutin | 0.8:1 | 92 | 18.01 |
| Diosmetin | 0.8:1 | 148 | 1.74 |
| Naringenin | 0.8:1 | 148 | 0.98 |

Note: IT = Inlet temperature; M:E = maltodextrin gram ratio per 1 g of extract; *w/w* = weight/weight; * mg Gallic acid equivalent/100 g powder.

The effect of the temperature ($p > 0.05$) and maltodextrin ratio ($p < 0.05$) interaction on TPC can be seen in Figure 2a, where the maxima response is obtained for two different ranges. The first used a maltodextrin (DE20):Habanero pepper extract ratio above 2.5:1 with a temperature lower than 120 °C, and the second was achieved with a maltodextrin ratio lower than 1.5:1 *w/w* (M:E) and a temperature higher than 40 °C. In this way, the modeling had maxima and minima behaviors.

Protocatechuic acid (Figure 2c) as a variable dependent of the inlet temperature and ratio of maltodextrin for microencapsulation presented a similar behavior to TPC (Figure 2a), where the surface response plot had maximum(red color zone) and minimum (blue color zone) behaviors. Only the maltodextrin ratio presented an effect ($p < 0.05$) on the Protocatechuic acid and TPC microencapsulates.

Coumaric acid (Figure 2e), Rutin (Figure 2g), Diosmetin (Figure 2i), and Naringenin (Figure 2f) presented a plot of minima where the central zone (blue color) represented the lowest concentration values of microencapsulated metabolites and a zone of maximum response (red color) where the highest concentration values of microencapsulated metabolites were obtained.

For the concentration of Coumaric acid, the maltodextrin ratio ($p < 0.05$) and temperature ($p < 0.05$) showed effects on the microencapsulated Habanero pepper extract; in contrast, their interaction (*p*-value $> 0.05$) did not present effects. Meanwhile, for the concentration of Rutin, Diosmetin, and Naringenin, the interaction of both factors did show an effect ($p < 0.05$).

### 3.2.5. Capsaicinoids Content in the Microencapsulated

During the optimization of the spray-drying conditions of the Habanero pepper extract, microcapsules were analyzed to determine the Capsaicinoid content.

The concentrations of capsaicin, Dihydrocapsaicin, and total Capsaicinoids, as well as the pungency reported in Scoville Heat Units (SHU) of the Habanero pepper extract prior to the microencapsulation, are shown in Table 6.

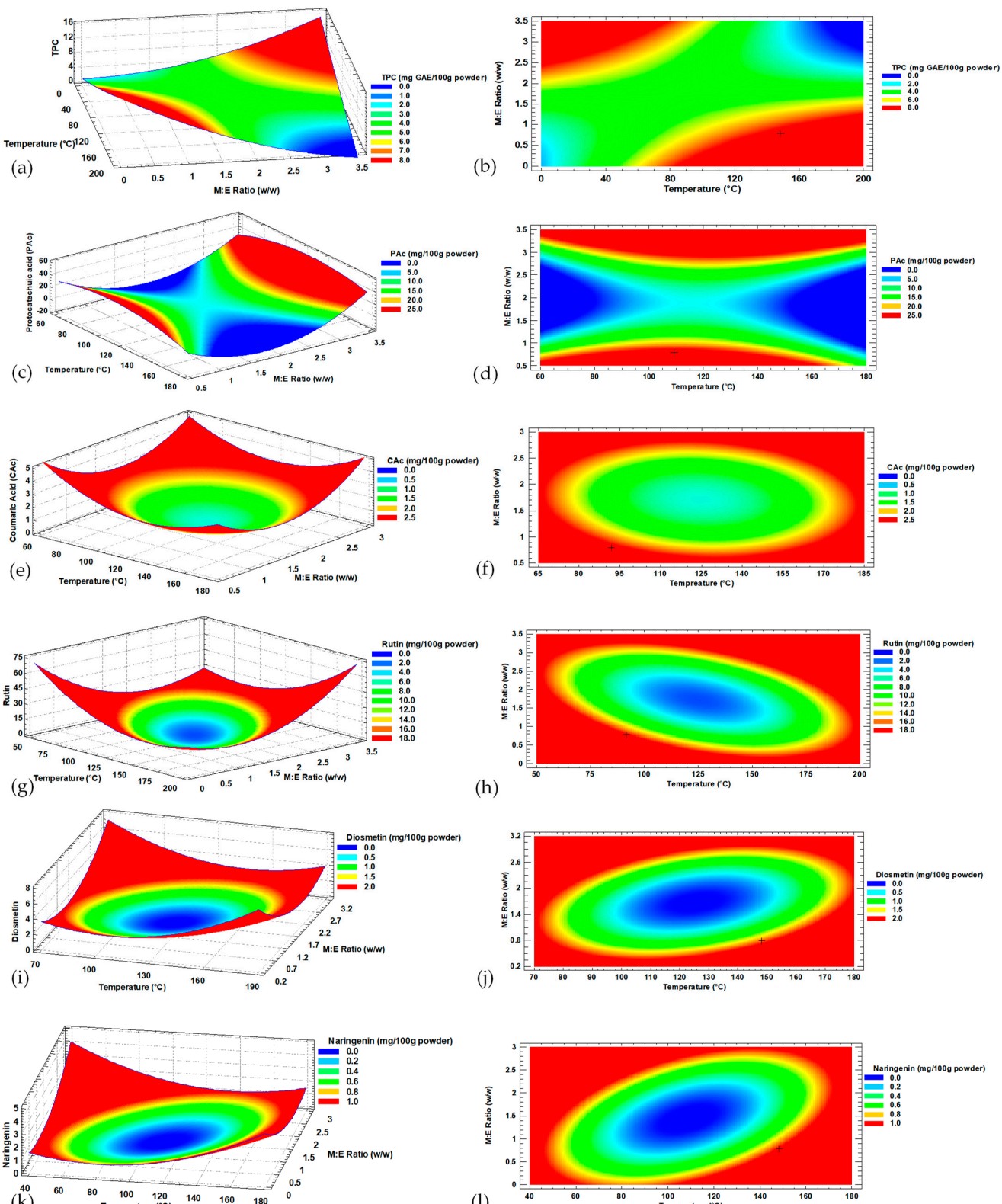

**Figure 2.** Response surface of (**a**) TPC = total polyphenol content; (**c**) Protocatechuic acid; (**e**) Coumaric acid; (**g**) Rutin; (**i**) Diosmetin; (**k**) Naringenin; and contour plots of (**b**) total phenolic content; (**d**) Protocatechuic acid; (**f**) Coumaric acid; (**h**) Rutin; (**j**) Diosmetin; and (**l**) Naringenin with factors of input inlet temperature and maltodextrin ratio; **+** indicates the response variable optimal value.

**Table 6.** Capsaicinoids content and Scoville Heat Units of Habanero pepper extract (*Capsicum chinense* Jacq.).

| Sample | CP * | DC * | TC * | SHU |
|---|---|---|---|---|
| Habanero pepper extract | $7.77 \pm 0.02$ | $1.18 \pm 0.02$ | $8.95 \pm 0.04$ | $134,280 \pm 38.00$ |

Note: CP = Capsaicin; DC = Dihydrocapsaicin; TC = total Capsaicinoids; SHU = Scoville Heat Units; values are means $\pm$ SD ($n = 3$); * mg/g dry matter.

According to the complete experimental design data (Table S3), concentrations of capsaicin and total Capsaicinoids were adjusted ($p < 0.05$) to a second-order model where only the latter presented an $R^2 > 70$ (Table 7). The prediction equations for each response variable were also obtained according to the regression coefficients of the multiple regression analysis obtained with ANOVA (Table S4).

**Table 7.** Results of multiple regression analysis of Habanero pepper extract Capsaicinoids microencapsulated by spray drying.

| Metabolite | *p*-Value | $R^2$ | Equation |
|---|---|---|---|
| CP | 0.0000 | 67.79 | $Y = -1.43601 + 0.0292452\ X_1 + 0.219263\ X_2 - 0.000132814\ X_1{}^2 + 0.000833333\ X_1\ X_2 - 0.142499\ X_2{}^2$ |
| TC | 0.0000 | 76.63 | $Y = -1.15088 + 0.0265272\ X_1 + 0.119397\ X_2 - 0.000122189\ X_1{}^2 - 0.117499\ X_1\ X_2 + 0.000916667\ X_2{}^2$ |

Note: CP = Capsaicin; TC = total Capsaicinoids; Y = metabolite mg/g powder; $X_1$ = inlet temperature (°C); $X_2$ = maltodextrin ratio per gram of extract.

Optimal conditions were established at 113.6 °C (IT) with a 1:1 $w/w$ ratio (M:E), according to canonical analysis, and predicted optimum values of PW of 0.35 mg/g and of 0.39 mg/g for capsaicin and total Capsaicinoids, respectively, at 112.1 °C (IT) with a 0.9:1 $w/w$ ratio (M:E).

The response surface and the contour plot (Figure 3) for the data obtained from microencapsulated capsaicin concentrations showed a behavior of maxima where the central zone in red color represented the maxima concentration of capsaicin; whereas, the blue zone showed the conditions by spray drying where the concentration would be minimal.

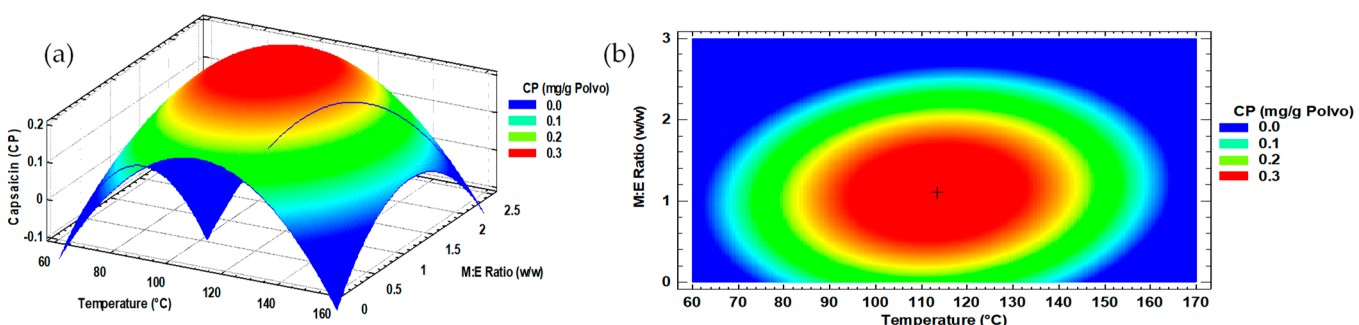

**Figure 3.** Capsaicinoids content response surface (**a**) and contour plot (**b**) with factors of input inlet temperature and maltodextrin ratio.

The behavior of the total Capsaicinoids content was similar to capsaicin (Figure S3).

The maltodextrin gram ratio per 1 g of Habanero pepper extract presented a significant effect ($p < 0.05$) on the concentration of microencapsulated capsaicin and total Capsaicinoids of the extract; meanwhile, the temperature showed an effect ($p < 0.05$) only on the first. Relatedly, the interaction of the main factors did not show an effect on both response variables (Table S4).

### 3.2.6. Model Validation for Antioxidant Capacity and Total Polyphenol Content

For the validation of the antioxidant capacity and the total content of polyphenols, mathematical models obtained by the response surface model, an inlet temperature of 148 °C, and a maltodextrin gram ratio per 1 g of Habanero pepper extract of 0.8:1 ($w/w$) were implemented (Table 8).

**Table 8.** Results of the validation experiment for antioxidant capacity and total polyphenol content; comparison with predicted values.

| Response Variable | Experimental * | Predicted * | Error Rate (%) |
|---|---|---|---|
| Antioxidant capacity | 38.84 ± 0.22% Inhibition | 33.27% Inhibition | 16.6 |
| TPC | 6.64 ± 0.08 mg GAE/100 g PW | 8.41 mg GAE/100 g PW | 21.04 |

Note: TPC = Total polyphenol content; GAE = Gallic acid equivalent; PW = powder; values are means ± SD ($n$ = 3); * microencapsulation condition 148 °C inlet temperature and 0.8:1 $w/w$ ratio (maltodextrin:Habanero pepper extract).

The validation experiment showed a value of antioxidant capacity of 38.84 ± 0.22% inhibition, where the predicted value was 33.27% inhibition; thus, the equation can be considered valid to predict this behavior of the microencapsulation process as a function of inlet temperature and the maltodextrin ratio for each gram of Habanero pepper extract.

Regarding the validation of the total polyphenol content, a value of 6.64 ± 0.08 mg GAE/100 g powder was obtained; whereas, the mathematical model predicted an 8.41 mg GAE/100 g powder value with an inlet temperature of 148 °C and a maltodextrin ratio of 0.8:1 ($w/w$) for each gram of Habanero pepper extract. An error rate of 21.04% was calculated.

Table S5 shows the polyphenol profile error rate from the validation experiment. Only Coumaric acid showed an error rate less than 25%; whereas, Protocatechuic acid, Rutin, Diosmetin, and Naringenin showed error rates above 40%. Therefore, only the model obtained for Coumaric acid was considered adequate for the prediction of the microencapsulated concentration by spray drying using maltodextrin as the encapsulating agent.

The mathematical model validation results obtained for the prediction of capsaicin and total Capsaicinoids concentration are shown in Table S6. In both cases, the error percentage was above 25%; thus, the prediction equations were not considered adequate for the prediction of the microencapsulation of these metabolites (Capsaicin and total Capsaicinoids) of a Habanero pepper extract by spray drying.

### 3.3. Physicochemical Characteristics of the Microencapsulated Habanero Pepper Extract

### 3.3.1. Microcapsules' Micrographs

Figure 4 shows micrographs of the shape of non-processed and spray-dried maltodextrin (DE20). Figure 4a represents particles with spherical shapes, with a smooth surface, and without cracks of maltodextrin (DE20) without the spray-drying process.

On the other hand, Figure 4b shows spray-dried maltodextrin (DE20) without the Habanero pepper extract. The morphology of the particles was spherical with a rough appearance, no cracks, and was smaller compared to the non-processed maltodextrin due to the presence of concavities on the surface.

The microcapsules obtained with a 1:1 ratio (maltodextrin: Habanero pepper extract, $w/w$) at a temperature of 100 °C (Figure 4c) had a less spherical and rough morphology; meanwhile, the microcapsules obtained with a 2:1 ratio (p/p) at a temperature of 140 °C (Figure 4d) showed spherical morphology and a greater number of concavities, presenting a rougher surface.

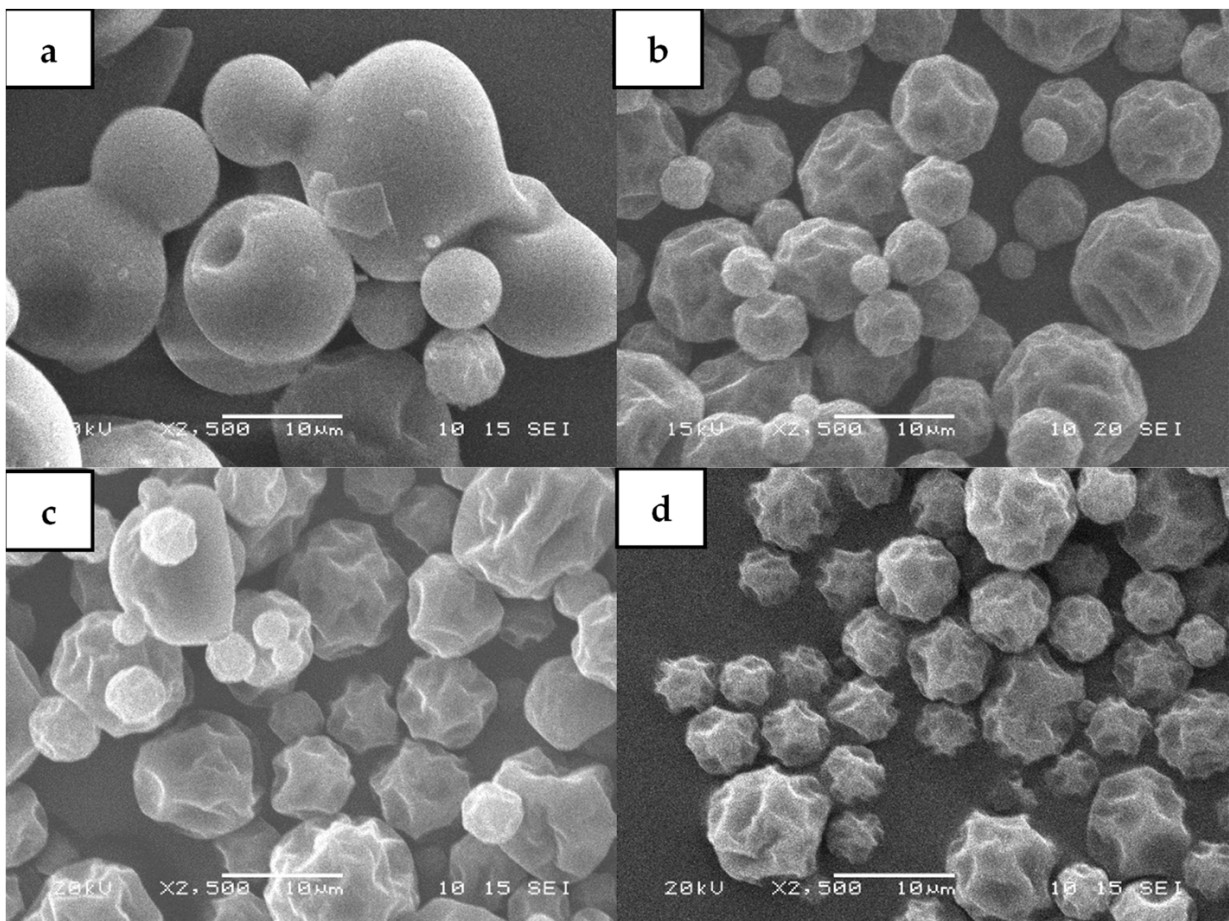

**Figure 4.** Microscopic images magnified at 2500×; (**a**) maltodextrin (DE20) without thermal process; (**b**) maltodextrin with thermal process at 140 °C; (**c**) microencapsulated Habanero pepper extract with ratio 1:1 *w/w* (M:E) at 100 °C; (**d**) 2:1 *w/w* ratio (M:E) at 140 °C.

### 3.3.2. FTIR Analysis of the Microencapsulated Habanero Pepper Extract

Extract and microencapsulate FTIR analysis showed characteristic peaks related to maltodextrin and phenolic compounds.

The infrared spectra of the extracts (Figure 5a) showed a broad band between 3368 and 3265 cm$^{-1}$ related to the hydroxyl groups from residuals of water molecules and polyphenols, in addition to absorptions at 2953 cm$^{-1}$, 2931 cm$^{-1}$, and 2855 cm$^{-1}$ related to the stretching of C–H bonds. Extracts also exhibited absorptions at 1635 cm$^{-1}$ (C=C ethylenic stretching) and 1517 cm$^{-1}$ assigned to phenyl ring bending. At 1451 cm$^{-1}$ and 1405 cm$^{-1}$, CH bending deformation in aromatic and aliphatic compounds was observed; meanwhile, C-O-C/C-O-H was observed as an intense peak at 1015 cm$^{-1}$. At 1280 cm$^{-1}$, phenyl C–H rocking vibrations were also observed. Many of these absorptions have been reported in cinnamon infusion where aromatic compounds with phenyl bonds, as those shown with polyphenols, were found [43].

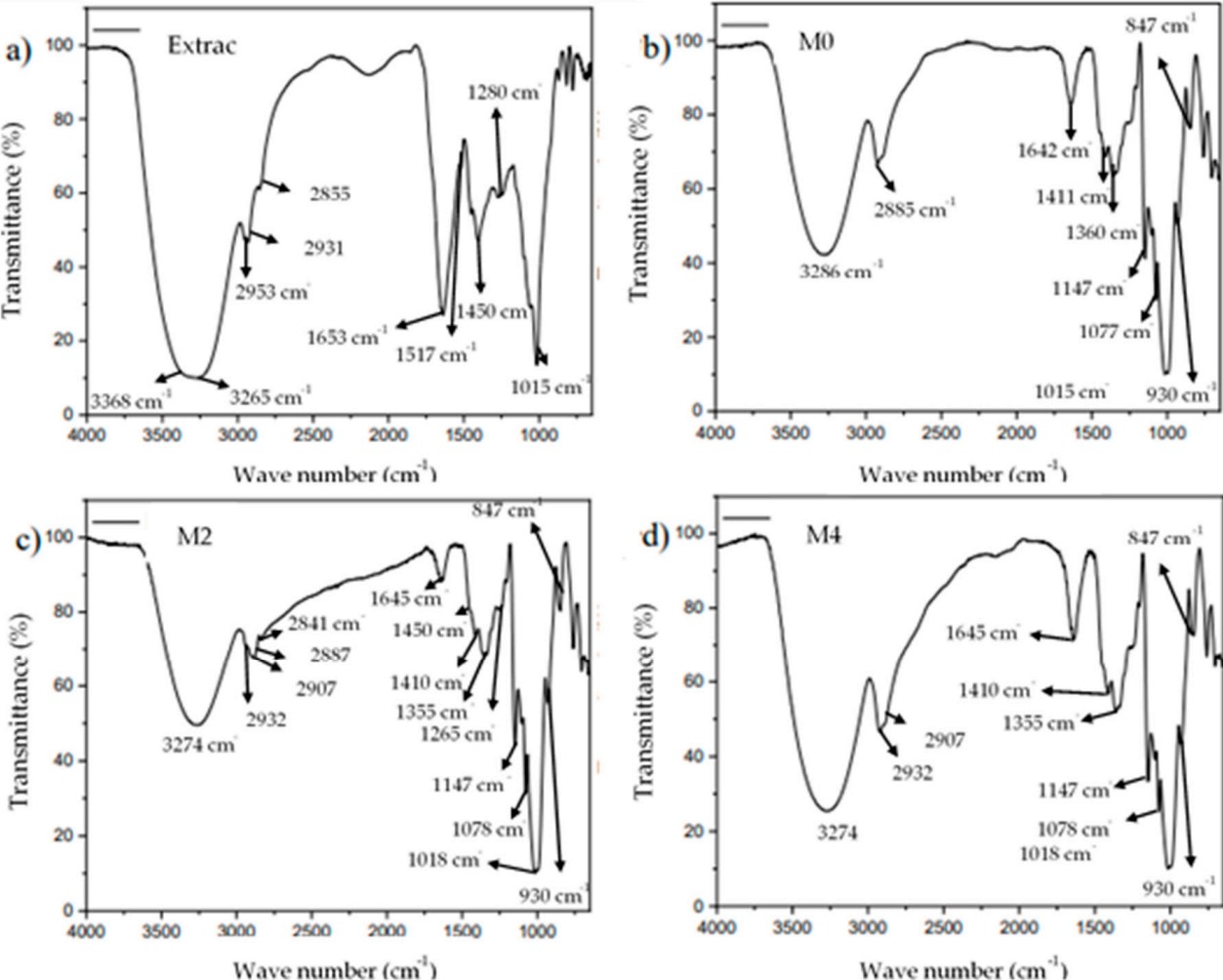

**Figure 5.** Infrared spectra of the Habanero pepper extract (**a**); sprayed maltodextrin at 140 °C without Habanero pepper extract, M0 (**b**); Habanero pepper extract microencapsulated at 140 °C with a 1:1 (*w/w*) ratio of maltodextrin:extract, M2 (**c**); and Habanero pepper extract microencapsulated at 140 °C with a 2:1 (*w/w*) ratio of maltodextrin:extract, M4 (**d**).

Pulverized maltodextrin (Figure 5b) showed similar absorptions with OH stretch peaks at 3286 cm$^{-1}$; whereas, at 2923 cm$^{-1}$ and 2885 cm$^{-1}$, absorptions related to CH$_2$ and CH stretching appeared. Pristine maltodextrin also exhibited a peak at 1642 cm$^{-1}$, which has been related to hydroxyl bending (in plane) and, by some authors, to free carboxylic groups [44]. In Figure 5b, at 1411 and 1360 cm$^{-1}$, bending CH$_2$ was observed. Finally, at 1147 cm$^{-1}$, 1077 cm$^{-1}$, and 1015 cm$^{-1}$, C-O-C stretching modes were observed.

Microencapsulates with high (M2) and low (M4) extract concentrations are shown in Figure 5c,d, respectively. In the M2 sample, infrared absorptions at 3274 cm$^{-1}$ corresponding to hydroxyl functional groups were observed; whereas, additional C–H absorptions were observed at 2932 cm$^{-1}$ (from extract), 2907, 2887 cm$^{-1}$ (from maltodextrin), and 2841 cm$^{-1}$. At 1645 cm$^{-1}$, a broad band was observed which can be related to both extracts and maltodextrin. Aromatic C-C vibrations were observed at 1450 cm$^{-1}$ as a small shoulder, and peaks at 1410 cm$^{-1}$ and 1355 cm$^{-1}$ from maltodextrin were also seen. Possible phenyl CH rocking from extracts was observed at 1265 cm$^{-1}$. Maltodextrin was notorious at 1147 cm$^{-1}$, 1078 cm$^{-1}$, and 1018 cm$^{-1}$, where the last peak also included those from the extracts.

Finally, at 930 cm$^{-1}$ and 847 cm$^{-1}$, absorptions corresponding to the maltodextrin were also observed. In M4 samples, similar peaks were observed except for a more intense absorption at 1645 cm$^{-1}$, indicating the higher maltodextrin:extract ratio used. Overall, these results confirmed that the spray-drying technique used in this study does not affect the structure of the maltodextrin and provided evidence of the presence of the extracts in agreement with previous studies [45].

### 3.3.3. Color of Microencapsulates of Habanero Pepper Extract

The color parameters L*, a*, b*, Hue°, and Chroma were analyzed for each microencapsulate resulting from the experimental design as well as in the microencapsulation obtained with the conditions of the mathematical model validation for antioxidant capacity and total polyphenol content. The results are shown in Table 9.

Microencapsulates corresponding to experiments four and twelve presented the highest luminosity. According to Li et al. [32], at a higher concentration of the encapsulating agent, an increase in the luminosity of the microencapsulates is observed. This agrees with the data obtained in the present work where the ratio of maltodextrin:extract was 2:1 and 2.2:1 *w/w* (M:E), respectively, in comparison to the microencapsulates from experiment one that presented the lower luminosity with a 1:1 *w/w* (maltodextrin:extract) ratio.

The color parameters a* and b* in the microencapsulates presented positive values, denoting a reddish (+a*) and yellowish (+b*) trend, respectively. According to Deveoğlu et al. [46], this phenomenon was explained due to the presence of metabolites of interest, such as phenolic compounds, mainly flavonoids (e.g., kaempferol, luteolin, etc.), flavones (e.g., hesperidin), and flavonols (e.g., quercetin), that were also identified (Table S7) in both the extract and microencapsulates of the Habanero pepper (*Capsicum chinense* Jacq.). On the other hand, the parameter a* presented a small tendency towards positive values; thus, the absence of a reddish color in the microencapsulates was also an indicator of a low concentration of anthocyanins, phenolic compounds that provide a red color to the plants, fruits, and leaves. The low concentrations of anthocyanins in the microencapsulates could be due to their sensitivity to thermal processes and practically the absence in immature Habanero peppers [47,48].

**Table 9.** Values of antioxidant capacity and total polyphenol content obtained from the $2^2$ CCD for the spray-drying condition optimization of Habanero pepper extract.

| Exp | Factors | | | | Response Variables | | | | | Graphic Representation ** |
|---|---|---|---|---|---|---|---|---|---|---|
| | Coded Values | | Real Values | | Color Parameters (CIELAB) | | | | | |
| | $X_1$ | $X_2$ | IT (°C) | E:M (*w/w*) | L* | a* | b* | Hue° | Chroma | |
| 1 | −1 | −1 | 100 | 1:1 | 82.83 ± 0.19 [a] | 1.67 ± 0.5 [g] | 19.23 ± 0.05 [j] | 1.48 ± 0.00 [a] | 19.31 ± 0.05 [j] | |
| 2 | 1 | −1 | 140 | 1:1 | 86.53 ± 0.21 [d] | 0.67 ± 0.5 [de] | 13.27 ± 0.05 [b] | 1.52 ± 0.00 [c] | 13.28 ± 0.05 [b] | |
| 3 | −1 | 1 | 100 | 2:1 | 87.67 ± 0.12 [f] | 0.63 ± 0.5 [cd] | 15.33 ± 0.12 [g] | 1.53 ± 0.00 [d] | 15.35 ± 0.13 [g] | |
| 4 | 1 | 1 | 140 | 2:1 | 88.40 ± 0.00 [h] | 0.50 ± 0.00 [b] | 12.13 ± 0.05 [a] | 1.53 ± 0.00 [d] | 12.14 ± 0.05 [a] | |
| 5 | 0 | 0 | 120 | 1.5:1 | 88.17 ± 0.12 [gh] | 0.57 ± 0.05 [bc] | 13.77 ± 0.05 [c] | 1.53 ± 0.00 [d] | 13.78 ± 0.05 [c] | |
| 6 | 0 | 0 | 120 | 1.5:1 | 87.07 ± 0.21 [e] | 0.73 ± 0.05 [e] | 14.80 ± 0.00 [f] | 1.52 ± 0.00 [c] | 14.82 ± 0.00 [f] | |

**Table 9.** *Cont.*

| Exp | Factors | | | | Response Variables | | | | | Graphic Representation ** |
| | Coded Values | | Real Values | | | Color Parameters (CIELAB) | | | | |
| | $X_1$ | $X_2$ | IT (°C) | E:M (*w/w*) | L* | a* | b* | Hue° | Chroma | |
| 7 | 0 | 0 | 120 | 1.5:1 | 87.40 ± 0.08 ef | 0.70 ± 0.00 de | 14.40 ± 0.08 d | 1.52 ± 0.00 c | 14.42 ± 0.08 d | |
| 8 | 0 | 0 | 120 | 1.5:1 | 87.73 ± 0.05 fg | 0.57 ± 0.05 bc | 14.30 ± 0.00 d | 1.53 ± 0.00 de | 14.31 ± 0.00 d | |
| 9 | −1.414 | 0 | 92 | 1.5:1 | 87.03 ± 0.66 e | 0.53 ± 0.05 b | 14.90 ± 0.08 f | 1.54 ± 0.00 ef | 14.91 ± 0.08 f | |
| 10 | 1.414 | 0 | 148 | 1.5:1 | 86.93 ± 0.12 de | 0.50 ± 0.00 b | 14.63 ± 0.09 e | 1.54 ± 0.00 f | 14.64 ± 0.09 e | |
| 11 | 0 | −1.414 | 120 | 0.8:1 | 85.03 ± 0.19 c | 0.97 ± 0.05 f | 17.70 ± 0.08 i | 1.52 ± 0.00 b | 17.73 ± 0.08 i | |
| 12 | 0 | 1.414 | 120 | 2.2:1 | 88.17 ± 0.05 gh | 0.33 ± 0.05 a | 13.20 ± 0.00 b | 1.55 ± 0.00 g | 13.20 ± 0.00 b | |
| 13 + | - | - | 148 | 0.8:1 | 83.93 ± 0.17 b | 1.00 ± 0.00 f | 17.53 ± 0.05 h | 1.51 ± 0.00 b | 17.56 ± 0.05 h | |

Note: Exp = Experiment; IT = inlet temperature; M:E = ratio of maltodextrin per gram of Habanero pepper extract; *w/w* = weight/weight; + validation experiment; values are means ± SD (*n* = 3). Different letters in the same column indicate a significant statistical difference; ** the CIELAB scale color parameters conversion to images was performed with the e-paint converter: https://www.e-paint.co.uk/convert-lab.asp (accessed on 20 November 2022).

### 3.3.4. Moisture of Microencapsulated Habanero Pepper Extract

The microencapsulation obtained at an inlet temperature of 140 °C with a maltodextrin ratio of 1:1 (*w/w*) presented the lowest moisture (2.07 ± 0.01%). In general, all the microencapsulates obtained (Table 10), with a temperature equal or greater than 140 °C, presented a low moisture content (<4%); whereas, the microencapsulation process under 100 °C and a 1:1 (*w/w*) maltodextrin ratio presented the highest moisture percentage (6.90 ± 0.40%). This behavior was described by Tolun et al. [38], where phenolic compounds extracted from a grape pomace were microencapsulated at different inlet temperatures and encapsulating agents (maltodextrin, arabic gum, etc.) ratios, finding that in microencapsulates (powders), the higher the temperature, the lower the moisture due to a greater water loss during spray drying.

Microencapsulates with 4% moisture or less have a positive prognosis in terms of shelf life due to a better stability of the structure of the microcapsule that efficiently protects the metabolites inside [49]. In this way, the conditions used for the microencapsulates with the aforementioned moisture percentage are viable to protect the polyphenols and antioxidant capacity of the Habanero pepper extract during storage.

**Table 10.** Microencapsulate moisture of the experimental design for the optimization of the spray-drying conditions of a Habanero pepper extract.

| Exp | Spray-Drying Conditions | | Moisture (%) |
| | IT (°C) | E:M (*w/w*) | |
|---|---|---|---|
| 1 | 100 | 1:1 | 6.90 ± 0.40 [f] |
| 2 | 140 | 1:1 | 2.07 ± 0.01 [a] |
| 3 | 100 | 2:1 | 5.91 ± 0.01 [e] |
| 4 | 140 | 2:1 | 2.53 ± 0.00 [ab] |
| 5 | 120 | 1.5:1 | 3.52 ± 0.02 [c] |
| 6 | 120 | 1.5:1 | 5.51 ± 0.12 [e] |
| 7 | 120 | 1.5:1 | 4.62 ± 0.31 [d] |
| 8 | 120 | 1.5:1 | 3.71 ± 0.00 [c] |
| 9 | 92 | 1.5:1 | 4.36 ± 0.06 [d] |
| 10 | 148 | 1.5:1 | 2.77 ± 0.14 [b] |
| 11 | 120 | 0.8:1 | 3.61 ± 0.09 [c] |
| 12 | 148 | 2.2:1 | 3.74 ± 0.16 [c] |
| 13 [+] | 148 | 0.8:1 | 3.75 ± 0.06 [c] |

Note: Exp = Experiment; IT = inlet temperature; M:E = maltodextrin ratio per gram of Habanero pepper extract; *w/w* = weight/weight; + validation experiment; values are means ± SD (*n* = 3). Different letters in the same column indicate a significant statistical difference.

## 4. Discussion

The microencapsulation of volatiles present in the aroma [23] and oleoresins, rich in capsaicin [21,50] and carotenoids [25], of *Capsicum* using spray-drying technology has been reported in the literature. The implementation of different encapsulating agents, such as sodium caseinate, acetylated wheat starch, whey protein, gum arabic, and maltodextrin, has been also conducted with the sole objective to assess the microencapsulation conditions for preservation and to increase shelf life of these compounds of interest. However, no information has been found on the optimization of the microencapsulation of metabolites such as polyphenols and their antioxidant capacity in *Capsicum* methanolic extracts.

In the present study, observations revealed that the antioxidant capacity, total polyphenol content, and concentration of microencapsulated individual polyphenols were lower when compared to those of the Habanero pepper extract. This could be due to the temperature-dependent nature of the spray-drying process wherein the presence of oxygen and light can promote the degradation of metabolites in a solution (comprising the encapsulating agent and extract), even if the mixture is exposed to these conditions for a short duration [25]. Despite this, any significant effect of temperature on the individual and total polyphenol content was not found as well as the antioxidant capacity of the microencapsulated extract obtained from the Habanero pepper (*Capsicum chinense* Jacq.). However, it has been reported that the drying temperature can affect the concentration of microencapsulated polyphenols derived from extracts of different fruits. For instance, when a grape pomace extract was microencapsulated using maltodextrin (DE20) as the encapsulating agent (a 1:1 *w/v* ratio) at inlet temperatures ranging from 120 °C to 160 °C, a decrease in the concentration of microencapsulated phenolic compounds was observed compared to the total phenolic compounds of the extract due to the increase in the drying temperature [38]. This trend was also observed in the work of Lingua et al. [49] wherein polyphenols from a blueberry extract were microencapsulated using maltodextrin (DE14.7) with an inlet temperature of 160 °C, resulting in a 20% loss. Therefore, spray drying can lead to the degradation of phenolic compounds due to drying temperatures, which can

decrease the antioxidant capacity of the encapsulation as these bioactive compounds are mainly responsible for this property in plant extracts [51].

The ratio of the encapsulating agent had a significant effect on the total polyphenol content, Coumaric acid, and antioxidant capacity of the microencapsulates of the Habanero pepper extract. This effect on the concentration of microencapsulated bioactive compounds has already been reported in other studies [33,38]. For example, a maltodextrin concentration twice greater in relation to the mass of the fruit extract (*Vitis vinifera* L. and *Manilkara zapota*) resulted in a 10 to 50% reduction in the total polyphenol concentration. Corrêa-Filho et al. [26] also reported that an increase in the encapsulating agent (gum arabic or inulin) led to a decrease of up to 70% in the antioxidant capacity of the microencapsulates compared to the antioxidant capacity of the extract when spray drying an ethanolic extract of *Solanum lycopersicum* L. The observed pattern can be attributed to the fact that encapsulating agents do not contribute to the antioxidant capacity, and the escalation in the ratio of encapsulating agent results in the dispersion of metabolites of the extract, owing to the increase in total solids concentration. [27].

Among the different optimized individual polyphenols analyzed in the microencapsulates obtained under optimal spray-drying conditions (IT = 148 °C, M:E = 0.8:1 $w/w$), Rutin presented an optimized concentration of $4.1 \pm 0.02$ mg/100 g PW. This metabolite belongs to the subfamily of glycosylated flavonols, which are widely distributed among plants and synthesized through the phenylpropanoid pathway [52]. Like many phenolic compounds, Rutin has a wide variety of biological properties. For example, it can promote apoptosis of human cancer cells, such as breast cancer (MDA-MB-231), leukemia (K562), and lung carcinoma, among others. It has also been found to eradicate colon tumor cells in animal models (SW480) [53,54]. Additionally, Rutin has been used as a treatment in rats with streptozotocin-induced diabetes, where the Rutin-treated group had lower blood glucose when compared to the untreated group [54] due to the antioxidant and anti-inflammatory capacities of this metabolite [55,56].

Recently, Rutin has been proven to be effective against SARS-CoV-2 by inhibiting its replication thanks to its affinity with the virus [57]. However, despite its wide variety of biological properties, this polyphenol has low bioavailability when consumed orally due to its low solubility. Therefore, encapsulation using polysaccharides, such as maltodextrin and β-cyclodextrin, is proposed due to the ability to develop hydrogen bonds with this phenolic compound, which improves its solubility and therefore its bioavailability in the small intestine. The shelf life is also increased due to the stabilization of the metabolite. This metabolite represents an option as an ingredient for the formulation of drugs or functional foods [53,56,58,59].

To use these microencapsulates as functional ingredients, the inclusion of the bioactive compound in the encapsulating agent must be confirmed as well as the presentation of certain physicochemical characteristics that allow the preservation of the metabolite of interest to benefit the final consumer from its bioactive properties [60,61]. In this work, the use of FTIR confirmed the presence of phenolic compounds in the extract as well as their inclusion in the microcapsules with observed peaks at 1600 and 1700 cm$^{-1}$ (C-C or C=C polyphenol bonds) and 3600 and 3200 cm$^{-1}$ (polyphenol-maltodextrin bonds, phenolic hydroxy groups). Ferreira et al. [50] studied these characteristic bands where phenolic compounds from Astrocaryum vulgare Mart. were microencapsulated with maltodextrin (DE10) under spray-drying conditions of 100 °C (IT), an injection flow of 7.5 mL/min, and a pressure of 6 bar. The FTIR spectra of the microcapsules showed a band at 3381 cm$^{-1}$, corresponding to the stretching vibrations of hydroxyl groups of phenolic compounds, and peaks at 1624 cm$^{-1}$ that confirmed the presence of carbonyls (C=O). These peaks (microcapsules) are similar to those of the encapsulating agent which overlapped the peaks found in the spectrum of the extract. Sarabandi et al. [62] also reported a band in the region defined between 2800 and 3700 cm$^{-1}$ due to the presence of hydrogen bonds formed by the phenolic compounds of the eggplant extract with maltodextrin (DE18-20) used as an encapsulating agent during spray drying (IT = 140 °C, feed flow 15 mL/min, 4.5 bar).

Peaks between 900 and 1600 cm$^{-1}$ related to phenolic compounds were also observed with bands at 1618 cm$^{-1}$, corresponding to chlorogenic acid (1600–1685 cm$^{-1}$), according to Monje et al. [63]. The same peak was found in both the Habanero pepper extract and the microencapsulates, which contained a high concentration of chlorogenic acid. In this way, the FTIR analysis is a useful tool to confirm the presence of phenolic compounds in Habanero pepper extracts and its binding (encapsulation) to maltodextrin.

Another relevant parameter is the microcapsules final moisture at the end of the spray-drying process as an important characteristic that could predict the stability and shelf life of the product (powder). Moisture is also an indicator of the drying process; in this way, an adequate moisture indicates that the inlet temperature selected for the spray-drying process was correct. Therefore, it is expected that final moisture will be less than 6% [61] to achieve a longer shelf life and fewer degradations of the encapsulated bioactive compounds. In this regard, the microcapsules obtained with the optimized conditions (IT = 148 °C; M:E = 0.8 *w/w*) presented a final moisture of 3.75 ± 0.06%, an adequate value. The moisture values of the microencapsulates are consistent with that reported by Lingua et al. [49], where the effect of maltodextrin (DE 14.7) as an encapsulating agent (20% and 30%) in an ethanolic extract of blueberry rich in polyphenols was assessed at different inlet temperatures (140 °C and 160 °C). It was found that as the temperature and concentration of the encapsulating agent increase, the moisture decreases thanks to a heat transfer increased during the atomization of the solution besides the increasing solids in the injected solution. In this study, a microcapsule final moisture below 4% was reported with an inlet temperature of 160 °C using 30% maltodextrin; whereas, the microcapsules obtained a final moisture close to 5% at 140 °C with the same percentage of encapsulating agent. Tolun et al. [38] also observed that the final moisture decreased with an increasing extract to maltodextrin ratio from 1:1 to 1:2 *v/v* if the temperatures were higher than 140 °C (160 °C and 180 °C) in microencapsulates where maltodextrin was used as the only encapsulating agent, compared to the final moisture in microcapsules with a 1:2 *v/v* ratio (Extract:maltodextrin) at temperatures of 120 °C and 140 °C, regardless of the dextrose equivalent value (DE4-7, DE17-20). Finally, the morphology of the microcapsules was not affected by temperature, although the sudden increase in temperature above the boiling point of water generated a rapid loss of moisture, causing a shrinkage and formation of a hollow structure with concavities in the surface. Due to the toothed morphology and concavities observed, an undesirable and greater difficulty for reconstitution could develop. This could be prevented by increasing the concentration of maltodextrin to avoid a rapid loss of water during spray drying, obtaining more spherical and smooth structures, making the microencapsulate suitable for the food industry, according to Zhang et al. [64].

## 5. Conclusions

The microencapsulation conditions, inlet temperature, and encapsulating agent ratio of the Habanero pepper extract showed an effect on the antioxidant capacity, concentrations of the individual and total polyphenols, as well as the concentrations of capsaicin and total Capsaicinoids of the microencapsulated Habanero pepper (*Capsicum chinense* Jacq.). The microencapsulates with the best microencapsulation conditions (IT = 148 °C; M:E = 0.8 *w/w*) presented the highest antioxidant capacity (38.84 ± 0.22% inhibition), a high concentration of polyphenols (6.64 ± 0.08 mg GAE/100 g PW), and the highest contraction of secondary metabolites, such as Chlorogenic acid (42.67 ± 9.63 mg/100 g PW), Coumaric acid (2.27 ± 0.10 mg/100 g PW), Cinnamic acid (2.61 ± 0.01 mg/100 g PW), Quercetin and Luteolin (1.56 ± 0.010 mg/100 g PW), and Ellagic acid (2.49 ± 0.02 mg/100 g PW); considering the hydroxyl groups in their structure that develop hydrogen bonds with maltodextrin, this all resulted in a high affinity. The microencapsulated Habanero pepper extract obtained in this work was optimized to obtain the best antioxidant capacity and the best content of bioactive compounds (optimized individual polyphenols), as well as its adequate physicochemical characteristics (moisture, shape, and color), providing this product the potential to be used as an ingredient for functional food development. It is

recommended to carry out more experiments using different combinations of encapsulating agents during the spray-drying process in order to compare and achieve a higher shelf life, antioxidant capacity, and/or microencapsulated polyphenol concentration and to assess the rate release of metabolites from microcapsules, as well as their bioactivity and bioavailability during this process.

**Supplementary Materials:** The following supporting information can be downloaded at: https://www.mdpi.com/article/10.3390/pr11041238/s1, Figure S1: Calibration curve for the determination of the total polyphenol content by the Folin–Ciocalteu methodology; Figure S2: Chromatograms of the polyphenol profile obtained by ultra-high pressure liquid chromatography (UPLC); A) 1. Gallic acid, 2. Protocatechuic acid, 3. Catechin, 4. Chlorogenic acid, 5. Cinnamic acid, 6. Rutin, 7. Quercetin and luteolin, 8. Kaempferol; B) 9. Vanillin, 10. Coumaric acid, 11. Ferulic acid, 12. Ellagic acid, 13. Naringenin, 14. Apigenin, 15. Diosmetin; C) 16. Vanillic Acid, 17. Diosmin and hesperidin, 18. Neohesperidin; Figure S3: Total Capsaicinoids content response surface (a) and contour plot (b) by factors of input inlet temperature and maltodextrin ratio. Table S1: ANOVA result of the antioxidant capacity first-order model; Table S2: ANOVA result of the antioxidant capacity second-order model; Table S3: Results of the microencapsulated Capsaicinoids of the complete experimental design for the optimization of the spray-drying conditions of a Habanero pepper extract; Table S4: Model for capsaicin and total Capsaicinoids as function input variables and analysis of variance; Table S5: Results of the validation experiment for the microencapsulated polyphenol profile and their comparison with the predicted values of the mathematical model; Table S6: Results of the validation experiment for capsaicin and total Capsaicinoids content and their comparison with the predicted values of the mathematical model; Table S7: Polyphenol profile of the Habanero pepper (*Capsicum chinense* Jacq.) extract and microencapsulates by spray drying.

**Author Contributions:** Conceptualization, I.M.R.-B.; methodology, K.A.A.-B.; software, K.A.A.-B.; validation, I.M.R.-B., M.O.R.-S. and J.V.C.-R.; formal analysis, I.M.R.-B.; investigation, I.M.R.-B., K.A.A.-B., M.O.R.-S. and J.V.C.-R.; resources, I.M.R.-B.; data curation, I.M.R.-B.; writing—original draft preparation, K.A.A.-B.; writing—review and editing, I.M.R.-B., M.O.R.-S. and J.V.C.-R.; visualization, I.M.R.-B.; supervision, I.M.R.-B.; project administration, I.M.R.-B.; funding acquisition, I.M.R.-B. All authors have read and agreed to the published version of the manuscript.

**Funding:** This research was funded by the National Council of Science and Technology of Mexico (CONACYT), which financed project No. 257588 and scholarship 661099 for Kevin Alejandro Avilés-Betanzos.

**Data Availability Statement:** Not applicable.

**Conflicts of Interest:** The authors declare no conflict of interest.

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
