# Peer review of "Optimization of Spray-Drying Conditions of Microencapsulated Habanero Pepper (Capsicum chinense Jacq.) Extracts and Physicochemical Characterization of the Microcapsules"

_processes, doi:10.3390/pr11041238_

Round 1

Reviewer 1 Report

This article "Optimization of spray-drying conditions for Habanero pepper’s (Capsicum chinense Jacq.) extract microencapsulation and its physicochemical characterization” However, this is not a novel subject but it can be almost suitable for processes, though there are still some issues that should be amended.

Title: Please use a more appropriate and concise title.

Abstract: The type of statistical design used in this research should be mentioned.

Keywords: They were well chosen

Introduction:

In the introduction, it is not well explained why ultrasound pretreatment is used and what is its superiority over other methods?

Also, why is this method used for encapsulation despite the fact that there are more updated and newer methods, but what is its superiority over other methods?

The review of sources has not been done well and the gap of this research has not been well explained compared to other studies. Relevant sources were recommended https://doi.org/10.1556/066.2020.49.3.11 ; https://doi.org/10.1186/s40538-021-00220-z

Materials:

Please write materials as the Company Name (city, country), especially for chemical analysis assessment which was used in the study.

Methodology:

Why is the Moisture test mentioned twice, state both together?

Colorimetry should be expressed based on ΔE*

 Results:

In line 274 Why did the authors only state the Moisture? Then other physicochemical tests???

The order of conducting the tests is not correct and does not follow a specific process at all, which confuses the reader.

Discussion: Discussion must improve and, in some cases, it is weak.

Conclusions: It's better to try to make it more scientific, comprehensive and concise in detail, especially.

References: in some cases, if possible, provide newer references.

Reviewer 2 Report

Authors of manuscript determined the best microencapsulation conditions  for Habanero pepper's extract, thanks to which microencapsules exhibited the highest antioxidant capacity. Authors determined also the content of different polyphenols in extract and the physicochemical characteristics of microencapsulates.

It is interesting work. I have only a few comments:

From line 98 is a wrong numbering of subsections, please improve it.

line 165: it should be "Na2CO3" instead of "NaCO3"

line 227: should be "DPPHadj" instead of "DDPHadj"

line 650: should be "SARS-CoV-2"

line 697: should be "(140 °C and 160 °C)"

In this manuscript Authors determined the antioxidant capacity, it would be also interesting to evaluate antioxidant activity and expressing it as the IC50 value for the microencapsulated extract, and also comparing this IC50 value with control, e.g. vitamin C.

Reviewer 3 Report

Comments and suggestions

Ms. No.: processes-2294921 

Title: Optimization of spray-drying conditions for Habanero pepper’s
(Capsicum chinense Jacq.) extract microencapsulation and its physicochemical
characterization

Capsicum chinense Jacq. is worldwide recognized for its unique organoleptic characteristics, as well as its capsaicin content; however, other bioactive compounds of global importance such as phenolic compounds with antioxidant capacity have also been extracted. Besides, the extracts could be obtained by green extraction methods that present a high sensitivity to environmental conditions. Also, a viable option to avoid the degradation of bioactive compounds and to leverage their properties is microencapsulation by spray drying, considering factors such as inlet temperature (IT) and maltodextrin:extract ratio (M:E). Thus, the objective of this work was to establish the optimal spray drying conditions (IT and M:E) of a Habanero pepper extract with a final spray dried product characterization. Results showed that the optimal spray drying conditions were an IT = 148°C with an M:E = 0.8:1 w/w, where the antioxidant capacity (38.84 ± 0.22% Inhibition), total polyphenol content (6.64 ± 0.08 mg Gallic acid equivalent /100g Powder) and several individual polyphenols such as Protocatechuic acid, Coumaric acid, Rutin, Diosmetin, and Naringenin were evaluated. The microcapsules showed a spherical shape with concavities, moisture less than 5%; the inclusion of bioactive compounds was confirmed by UPLC and FTIR. The final dried product has the potential to be used as an ingredient for functional food development.

Some comments about this paper:

1)  Some quantitative findings might be mentioned in the Abstract section.

2)  Extended the Introduction section.

3)  Introduction; using biopolymer; stability of the materials via zeta potential measurements and swelling behavior, and other methods of modelling such as neural networks might be mentioned. The papers below clarifying these good for recommending

·        10.1007/s11130-022-00995-y

·        10.1016/j.indcrop.2022.116094

·        10.1016/j.foodchem.2006.12.067

4)  What is the novelty of the section 2.4?

5)  How did the authors find these ranges in the Table 1?

6)  What is the physical meaning of the Equation 5? Clarify

7)  Figure 1; how did the authors find the optimal conditions? Clarify

8)  What is the point that might be obtained from Table 5? Clarify

9)  Table 8; the error amounts are high, why?

10)              Figure 5; What is/are the points that the authors that obtained from Figure 5? The sizes of pellets might be mentioned.

11)              The optimization might be performed in a systematic manner such as used the classical of evolutionary methods. clarify

12)              Discuss physically on the obtained results.

Concluding, the paper definitely suited for publication with revision of the paper (major revision).

Round 2

Reviewer 1 Report

The manuscript was improved and this version can be accepted. 

Reviewer 3 Report

Revisions well done. Accept